# Relationship between Oat Consumption, Gut Microbiota Modulation, and Short-Chain Fatty Acid Synthesis: An Integrative Review

**DOI:** 10.3390/nu15163534

**Published:** 2023-08-11

**Authors:** Giovanna Alexandre Fabiano, Leila Marie Shinn, Adriane Elisabete Costa Antunes

**Affiliations:** 1School of Applied Sciences (FCA), State University of Campinas, 1300 Pedro Zaccaria St., Limeira 13484-350, SP, Brazil; giovanna.af97@gmail.com; 2Life Sciences, PepsiCo R&D, Chicago, IL 60607-0433, USA

**Keywords:** functional food, prebiotic, diet, β-glucan, metabolites, microbiome

## Abstract

The gut microbiota consists of a set of microorganisms that colonizes the intestine and ferment fibers, among other nutrients, from the host’s diet. A healthy gut microbiota, colonized mainly by beneficial microorganisms, has a positive effect on digestion and plays a role in disease prevention. However, dysregulation of the gut microbiota can contribute to various diseases. The nutrition of the host plays an important role in determining the composition of the gut microbiota. A healthy diet, rich in fiber, can beneficially modulate the gut microbiota. In this sense, oats are a source of both soluble and insoluble fiber. Oats are considered a functional ingredient with prebiotic potential and contain plant proteins, unsaturated fats, and antioxidant compounds. The impact of oat consumption on the gut microbiota is still emerging. Associations between oat consumption and the abundance of *Akkermansia muciniphila*, *Roseburia*, *Lactobacillus*, *Bifidobacterium*, and *Faecalibacterium prausnitzii* have already been observed. Therefore, this integrative review summarizes the findings from studies on the relationship between oat consumption, the gut microbiota, and the metabolites, mainly short-chain fatty acids, it produces.

## 1. Introduction

Oats (*Avena sativa* L.) contain soluble fibers such as β-glucan but also provide insoluble fibers such as lignin and cellulose [1]. Oats can be considered a functional food with prebiotic potential due to their fiber, lipid, and phenolic compound content [2]. Specifically, the prebiotic potential of β-glucan has been evaluated in several studies [3,4,5,6]. Of note, β-glucan has been associated with hypoglycemic and cholesterol-lowering effects [7]. More recently, it has been reported that β-glucan is fermented by the human gut microbiota, potentially impacting the composition of the microbiota while producing short-chain fatty acids (SCFAs), acetate, propionate, and butyrate [8]. Furthermore, oats have unique phenolic compounds, including avenacolysates, avenacins, and avenanthramides [9]. Some studies suggest that these phenolics compounds possess antioxidant and anti-inflammatory properties, and may provide protection against coronary heart disease, colon cancer, and skin irritation [10]. In addition, oats also provide proteins, including globulins such as avenalins, in addition to prolamins (avenins) [11].

On their own, oats do not contain gluten and can be consumed by patients with celiac disease [11]. Gluten is a protein found in some grains, such as wheat, barley, and rye [12]. Celiac disease is a systemic gastrointestinal disorder mediated by the immune system and induced by gluten ingestion [13]. While wheat and barley contain gliadin, oat changes this prolamin’s form to avenin, which has a molecular structure homologous to gliadin but is not identical [11]. While oats themselves are gluten-free, they are often cross-contaminated with other grains during processing [14]. Thus, gluten-free oats require controlled, contamination-free cultivation, processing, preparation, and storage [13]. The “Codex Alimentarius” of 2008 established 20 mg/kg as the maximum limit for the presence of gluten in “gluten-free” foods [15]. Most patients with celiac disease tolerate a minimum daily amount of 10 mg gluten, but it depends on the sensitivity of each individual [16]. Different reviews and meta-analyses concluded that a moderate consumption of oats is not associated with symptoms and immunological profile characteristics of patients with celiac disease [17,18,19]. However, the consumption of oats is only recommended for patients with celiac disease in remission with monitoring and caution [20]. Of note, although pure oats are safe for most people with celiac disease, adverse immunological reactions may occur [20,21,22].

Diet is one factor most strongly associated with gut microbiota modulation, if not the most associated. The gut microbiota is involved in several bodily functions, including nutrient digestion, metabolism, and fermentation, as well as endocrine and immunomodulatory functions [23,24,25]. The gut microbiota supports barrier function, improves intestinal permeability, and promotes mucous layer integrity [23]. Further, through dietary fiber fermentation, the gut microbiota produces gases (hydrogen and carbon dioxide), SCFAs, branched-chain fatty acids, and some organic acids and alcohols [24].

Due to its complexity and interindividual variation, there is a lack of definition of a “healthy gut microbiota” [25]. Further, the taxonomic composition of gut microbes among healthy individuals is diverse. Thus, identifying a specific set of microbes comprising universally “healthy” microbiota is unlikely. However, it is logical to conclude that a “healthy” gut microbiota would support the absence of any overt disease within the individual. Thus, a microbiome that resists stress and perturbation and can recover to an optimal functional profile is a more promising functional way to define a “healthy” gut microbiome [26]. Nonetheless, some situations can impair the microbiota and generate dysbiosis, which has been associated with several disease states, including inflammatory bowel disease (IBD), obesity, diabetes, allergies, and immune disease [27,28]. From a taxonomic composition standpoint, IBD has been associated with a reduction of Bacteroidetes and Firmicutes phyla, as well as an increase in Proteobacteria and *Escherichia coli* [27,29,30,31]. The transplantation of gut microbiota from obese mice to lean, germ-free mice increased intestinal permeability, lipogenesis, and adipogenesis [23,32]. In fatty liver diseases, low levels of Firmicutes and Bacteroides have been reported, in addition to high *E. coli* levels [28]. In individuals with Type 2 diabetes, a decrease in the phylum Firmicutes and Clostridia class was observed compared to healthy individuals [33]. Additionally, *Bacteroides fragilis*, *Enterococcus faecalis*, *Helicobacter hepaticus*, and *Fusobacterium nucleatum* have been associated with intestinal carcinogenesis [34,35,36], while an increased abundance of *Akkermansia muciniphila* has been observed in cancer patients who responded well to treatment [37].

The emerging connection of the gut to several other body systems (the gut–brain, gut–lung, and gut–skin axis) reveals a potential impact on health and quality of life. The gut microbiota connection axes have been investigated in several disease states, such as tuberculosis, psoriasis, and Parkinson’s disease [27]. As for the gut–lung axis, there is evidence that individuals with asthma have a higher expression of bacterial histamine decarboxylase in the gut, as well as increased abundance of the histamine-secreting species, *Morganella morganii* [27]. Gut–brain axis communication is proposed to occur through the enteric nervous system, including the sympathetic and parasympathetic systems [27]. Emerging data in cross-sectional clinical trials suggest the gut microbiota may be associated with anxiety, stress, depression, and cognitive learning behaviors [38].

The gut microbiota is impacted by age, genetics, exercise, antibiotics, smoking, and, especially, diet [28]. Dietary nutrients interact directly with the gut microbiota, affecting its composition and growth kinetics [39]. Since certain fibers can act as prebiotics or substrates that are selectively utilized by host microorganisms conferring a health benefit, examining the impact of including fiber-containing foods, such as oats, barley, chia, flaxseed, and soy, in the diet to modulate the gut microbiota has been a topic of continued research interest [40]. The fiber content of the grains varies according to the species, part of the plant, location, and characteristics of cultivation and processing [41]. Barley and oats have a high fiber content, around 10 to 15% [42]. A high-fiber diet is associated with greater microbial diversity [43]. However, a low-fiber diet was associated with an increase in enteric pathogenic microorganisms and a reduction in beneficial Firmicutes, which are responsible for metabolizing plant-derived polysaccharides into SCFAs [43]. As an example of a substrate for gut microbiota fermentation, soluble fiber from oats has been shown to be an energy source for butyrate-producing bacteria [44].

Although most of the scientific literature has focused on the effects of oat consumption on cardiovascular health, the present work proposes to focus on the benefits of oat consumption for promoting gut health, proper balance of the colon microbiota, and production of SCFAs by this complex community.

## 2. Methodology

The methodological approach utilized in this work was based on Kutcher and LeBaron’s guide on completing an integrative review [45]. The criteria for inclusion were the selection of original articles (clinical trials, in vitro, and in vivo trials) from 2012 to 2023. All relevant papers on the modulation of the gut microbiota with oat products or supplements were downloaded from Scopus and PubMed. Terms identified comprised three cores: the food matrix “oat,” the main interest “microbiota,” and the metabolites that represent an important relationship between oats and the gut microbiota, “SCFA—short-chain fatty acids.” Using the identified terms, an advanced search procedure was performed as follows: “oat” AND “microbiota” AND “short-chain fatty acids” AND “SCFA” on October and November 2022. This procedure was repeated in April 2023. From this advanced search, 58 studies were identified, including 35 from Scopus and 23 from PubMed (Figure 1).

Screening of publications was performed by reviewing the title, abstract, and keywords to determine potentially relevant studies. After screening, 45 articles were identified and duplicates were removed, leaving 31 publications for full text review. During full-text review, 15 publications were excluded because they were either reviews, not specific to oats, or the study was in a diseased population. Following full text review, 16 publications were included in the review.

Table 1 includes in vivo studies using animals, Table 2 includes in vitro studies, and Table 3 includes clinical studies in humans.

## 3. Oat as a Functional Food

According to the Food and Nutrition Board of the Institute of Medicine, the adequate intake (AI) for fiber is 14 g of total fiber per 1000 kcal, or 25 g for adult women and 38 g for adult men [61]. The AI for fiber represents the median fiber intake level observed to achieve the lowest risk of coronary heart disease [61]. While there is no tolerable upper-intake level for fiber, an excessive intake of dietary fiber can cause flatulence, bloating, and diarrhea. Therefore, consuming dietary fiber in adequate amounts can be beneficial to health, reducing the risk of several chronic diseases and supporting digestive health through modulation of laxation, fermentation, and the gut microbiota [62]. In this sense, oat intake can play a role in achieving the AI for fiber. The nutritional composition of oats includes carbohydrates, soluble fiber, proteins, lipids (such as unsaturated fatty acids), vitamins, minerals, and phenolic compounds. The International Food Information Council (IFIC) defines the term “functional foods” as foods or food components that provide a health benefit beyond basic nutrition [63]. Oats could be considered a functional food because of the presence of β-glucan, a soluble fiber shown to reduce cholesterol [63,64]. While there is no regulatory definition of functional food, the US Food and Drug Administration (FDA) authorized the use of a health claim regarding soluble oat fiber and the reduction in coronary heart disease risk based primarily on its cholesterol-lowering abilities [65,66]. The soluble fiber in oats may also influence gut microbiota.

Oats also have high protein levels compared to other cereals [67]. The proteins present in oats are mainly avenins and avenalins [11]. Oat proteins are of a higher quality than many other cereal grains since they have higher levels of essential amino acids, such as lysine [67]. Thus, oats may be a complementary alternative to animal protein due to the current market demand for non-animal protein substitutes. Bioactive peptides from oat protein have been studied and identified as compounds potentially beneficial to health in in vitro and animal studies [67]. Some of the benefits under investigation include reducing blood pressure and cholesterol, immunomodulation, and blood glucose control [67]. However, more clinical studies in humans are needed, as well as a better understanding of effective doses and potential mechanisms of action specific to oat protein.

Other bioactive compounds from oat include Vitamin E, flavonoids, phenolics, phytosterols, and avenanthramides [68]. These metabolites act as defense mechanisms during plant growth and have antioxidant functions [68,69]. Vitamin E, or tocopherol isomers, has a high antioxidant capacity and may also have anti-inflammatory properties, which could impact the risk of cancers and cardiovascular disease [70]. Phenolic compounds found in oats include caffeic acid, coumaric acid, phytic acid, gallic acid, and vanillic acid [68]. Avenanthramides (AVAs) are the most abundant phenolic alkaloids unique to oats [69,71]. There are more than 25 forms of AVAs described in oats, which have an antioxidant activity 30 times greater than other phenolic compounds [71,72]. AVAs have also been investigated for their anti-inflammatory, antiproliferative, pro-apoptotic and antiatherogenic actions [68,69,71].

### 3.1. Anticholesterolemic, Hypoglycemic, and Antihypertensive Properties

There is robust evidence that the consumption of oats or oat-containing foods lowers total cholesterol and low-density lipoprotein (LDL) cholesterol in healthy adults who are overweight or obese, as well as individuals with Type 2 diabetes [73,74,75]. A dose-response meta-analysis concluded that 3 g of β-glucan per day were able to effectively reduce total cholesterol [76]. One potential mechanism by which β-glucan modulates cholesterol metabolism is by increasing viscosity in the intestine, thereby influencing bile acid metabolism [64]. Increasing the viscosity of the intestinal contents limits bile acid reabsorption in the terminal ileum and increases their fecal excretion. In the absence of bile acid recycling, de novo synthesis of bile acids is required, resulting in the utilization of cholesterol [64,77]. Animal and human studies have demonstrated fecal excretion of bile acids and an elevation of bile acid synthesis after consumption of oats or isolated β-glucan [78,79]. Oats and isolated β-glucan may also modulate gut microbiota composition and function to influence cholesterol metabolism. Further, bile acids can be deconjugated by certain microbiota, which inhibits their reabsorption [67]. These microbes include certain *Bifidobacterium*, *Bacteroides*, and *Lactobacillus* species that have high activities of bile salt hydrolase, the enzyme responsible for the deconjugation of bile acids [64,80]. β-glucan has been shown to increase these microbes in animal and in vitro studies [64]. In addition, the proteins and lipids present in oats may also contribute to cholesterol control [81]. Higher concentrations of proteins and lipids in oats were responsible for lower serum levels of total cholesterol and LDL-cholesterol in rats fed with a hypocholesterolemic diet [81]. Health benefits are also attributed to phytochemicals present in oats. Antioxidant compounds, plant sterols, Vitamin E, and polyunsaturated acids are associated with the prevention of cardiovascular disease [82].

The viscosity created in the intestine by oat β-glucan may also contribute to the reduction of blood glucose [82]. The viscous nature of the bolus slows transit and makes it difficult for digestive enzymes to access carbohydrate molecules [83]. It also limits the access of monosaccharides to the luminal surface to be absorbed. Oat β-glucan has also been shown to promote satiety, possibly contributing to weight control [53,71]. A randomized controlled trial in adults with hypertension demonstrated a reduction in blood pressure in participants following a DASH diet with oat bran supplementation [84]. The participants were also able to reduce hypertension medications and experienced changes in the gut microbiota (increased abundance of *Bifidobacterium* and *Spirillum* populations) [84]. While further research is needed, there is emerging data that SCFAs, produced by the fermentation of fiber by gut microbiota, may activate receptors in the kidneys and blood vessels to inhibit renin release and decrease blood pressure [85]. Further, in animal and in vitro models, oat β-glucan has demonstrated immunomodulatory, anti-inflammatory, and antioxidant effects, but these need further investigation in human clinical trials [54,86].

### 3.2. Prebiotic Potential

The International Scientific Association for Probiotics and Prebiotics (ISAPP) defines a “prebiotic” as a substrate that is selectively used by host microorganisms and confers health benefits [40]. Prebiotics are fermented by the gut microbiota and selectively favor the growth of beneficial microorganisms [87,88]. While the ISAPP definition is the most widely used in scientific literature, few global regulatory agencies have defined prebiotics for use in foods or for claims. The FDA has not established a regulatory definition for prebiotics, but they are regulated similar to other food ingredients, such as food additives, or generally recognized as safe (GRAS) ingredients [89]. The European Food Safety Authority (EFSA) uses the FAO 2008 definition, which describes prebiotics as “non-viable food components that confer a health benefit to the host associated with modulation of the microbiota” [89,90]. Importantly, EFSA has noted that changes in microbiota alone are not a benefit, but the change in microbiota must result in a health benefit to the host [91,92]. Japan does not have a regulatory definition for prebiotic but does consider oligosaccharides and dietary fiber as foods that modify gastrointestinal conditions; it can be used for health-promotion purposes [93]. The Canada Food Inspection Agency considers the use of the prebiotic term on food labels to be an implied health claim, and a health claim can only be used when a physiological benefit is demonstrated in humans [94]. Thus, all definitions of a prebiotic include the mention of health promotion of the host through modulation of the gut microbiota [91]. Specific prebiotic benefits have not been defined but could include promoting the integrity of the intestinal mucosal barrier, improving nutrient digestion and absorption, increasing immunity, reducing pH and production of SCFAs, as well as inhibiting pathogenic microorganisms [43]. However, the challenge in meeting some regulatory definitions of prebiotics is proving these benefits result from the changes in the gut microbiome populations or function.

Figure 2 represents a summary of findings from the articles selected for this review. Therefore, the figure shows the performance of oats as a functional food, the modulation of the gut microbiota, and the production of SCFAs. Oat β-glucan has demonstrated potential prebiotic properties through its fermentation to SCFAs, but additional research is needed to demonstrate a health benefit of SCFAs. β-glucan has also been shown in human and animal studies to promote the growth of certain beneficial gut microbes. A randomized controlled trial reported a significant increase in *Bifidobacterium* and *Akkermansia*, as well as a reduction in the Sutterellaceae family when 80 g/d of oats were consumed as part of a normal diet for 45 days [59]. Other animal and in vitro studies reported an increase in *Bifidobacterium*, as well as other beneficial bacteria such as *Lactobacillus*, the Firmicutes phylum, and the Eubacteriaceae family [46,48,51,52,53,56]. Kristek et al. (2019) concluded that oat bran had a greater impact on the intestinal microbiota than supplementation with β-glucan or isolated polyphenols, suggesting the importance of food form [58]. Oat bran increased the proliferation of *Bifidobacterium adolescentis* after in vitro fermentation, but β-glucan alone did not [58]. Another in vitro study evaluated different ways of preparing oat bran, such as steaming, microwaving, and drying with hot air [55]. Preparation methods did not change the amount of dietary fiber or β-glucan, and all oat bran treatments were able to decrease the abundance of *Escherichia*–*Shigella* and increase *Faecalibacterium prausnitzii* in the gut microbiota [55]. However, steamed oat bran produced more SCFAs (acetic, propanoic, and butyric acid) than microwaved or hot air-dried oat bran [55]. This suggests heat treatment of oat or oat bran may increase its fermentability [95].

### 3.3. Short-Chain Fatty Acids and Related Metabolites

The gut microbiome has expansive metabolic capacity, and many different metabolites are produced as the microbes ferment undigested food. Short-chain fatty acids (SCFAs) are the most prevalent and well-researched metabolites produced during bacterial fermentation. A decrease in the abundance of SCFA-producing bacteria has been associated with chronic conditions, such as diabetes, inflammatory bowel disease, and atherosclerosis [96]. However, more clinical trials are needed to further elucidate the health effects of SCFAs and most plausible mechanisms for potential benefits.

Acetate, propionate, butyrate, isobutyrate, valerate, isovalerate, and hexanoate are the main SCFAs produced from dietary fibers in the human colon [23]. Acetate is produced at the greatest concentration by the intestinal microbiota [96]. Butyrate is produced at lower concentrations and is generated primarily by Firmicutes [97]. Propionate is produced by some Firmicutes, Bacteroidetes, and Verrucomicrobia, especially *Akkermansia muciniphila* [98,99,100]. Most acetate and propionate are absorbed into circulation, while butyrate is the primary energy source for intestinal epithelial cells [101,102].

SCFA-sensitive receptors are expressed in many cell types throughout the body, including immune cells, adipose tissue, cardiac tissue, skeletal muscle, and neurons [98,103]. Thus, SCFAs, particularly propionate and acetate, have the potential for broad action in the host, including metabolism, cell differentiation and proliferation, gene regulation, protein methylation, and phosphorylation [96,100]. In vitro and animal studies suggest SCFA production is associated with the synthesis and secretion of intestinal hormones, such as GLP-1 and PYY, which play a role in satiety, intestinal transit, and homeostasis [104,105]. In addition to being an energy source for epithelial cells, butyrate may also suppress cancer cell expansion [96]. Butyrate has also been considered as a therapeutic target for the treatment of Type 2 diabetes [106].

SCFAs have also been purported to have anti-inflammatory and immunomodulatory effects [107]. SCFAs have been shown to inhibit the expression of adhesion molecules, suppress macrophage and neutrophil recruitment, and decrease LPS-induced TNFα [108,109]. Butyrate has been shown in vitro to reduce intestinal permeability and stimulate mucus production to strengthen the epithelial barrier [96]. Butyrate can also modulate antimicrobial peptide secretion in the intestinal epithelium [96].

Oats may contribute to the production of SCFAs due to their fiber content, particularly β-glucan. Table 1 includes studies on oat and gut microbiota in animal models. In a study with rats fed a high-fat diet (HFD), oat supplementation increased butyrate concentrations in the intestine, which was associated with improved lipid metabolism, reduced oxidative stress, and attenuated inflammatory responses [46]. In a similar study, oat fiber helped attenuate the development of obesity and dyslipidemia caused by HFD in mice [47]. The authors suggested that the benefit of oat fiber may be due to SCFA production that could impact lipid and glucose metabolism [47]. This is supported by another study that reported a significant increase in intestinal production of acetate, propionate, and butyrate in hamsters fed HFD for 30 days [52]. The hamsters also experienced a decrease in total plasma and hepatic cholesterol, LDL-cholesterol, triglycerides, and an increase in fecal weight and bile acids [52]. In a study with oat flour and oat bran supplemented in HFD-fed rats, both products altered the gut microbiota composition and increased SCFAs concentrations, which was associated with decreased body weight, inflammation, and lipid levels [53]. Oat bran had promising effects on body weight and inflammation, significantly reducing adipocyte size and inflammatory TNF-α mRNA expression levels [53]. Finally, a diet containing a mixture of coarse cereals, including millet, maize, oat, soybean, and purple potato, reduced body weight gain and fat accumulation, improved glucose tolerance and serum lipid levels, and reduced systemic inflammation in mice treated with HFD [48]. These effects were associated with an increase in SCFA synthesis and an increase in the relative abundance of *Lactobacillus* and *Bifidobacterium* [48]. Intestinal integrity markers, including mucin and tight junction proteins, have also been associated with SCFA production in mice consuming oat bran [49]. A study investigating the gut–brain axis in an animal model of atherosclerosis reported 0.8% oat fiber supplementation for 14 weeks, which delayed the progression of cognitive impairment and increased expression of SCFA receptors and intestinal microbiota diversity [50]. The neuroprotective potential was suggested to result from the production of anti-inflammatory metabolites triggered by the fermentation of oat fiber to SCFAs [50].

In vitro studies have frequently been used to evaluate the impact of different dietary fibers and fiber treatments on the gut microbiota and subsequent metabolites. The findings of in vitro studies on oats are presented in Table 2. In one study evaluating different ways to process oat bran, steamed oat bran increased SCFA production more than microwave and hot-air oven processing [55]. When compared with other grains and fibers, such as corn, millet, and oligosaccharides in infant fecal inoculum, oat seems to preferentially increase *Lactobacillus* and promote butyrate production, even though overall SCFA production is similar among the grains [56]. However, other studies with adult fecal inoculum have reported increases in acetate and propionate and a decrease in butyrate with oat bran treatment, as well as increases in *Bifidobacterium* [103].

Table 3 details the two clinical trials reviewed on oat consumption and microbiota modulation. Valeur et al. (2016) evaluated 10 healthy subjects who ingested 60 g of oat porridge daily for one week [60]. Although the production of intestinal gas and excretion of SCFA did not change significantly, fecal levels of β-galactosidase and urease decreased after ingestion [60]. β-galactosidase is a microbial enzyme with activity similar to human lactase [60]. Thus, the authors hypothesized that the decrease in fecal levels of β-galactosidase reflects an adaptation of the microbiota not needing this enzyme for the digestion of oat porridge [60]. Furthermore, bacterial urease is an enzyme responsible for the hydrolysis of urea into ammonia [110]. In certain cases, high levels of ammonia are associated with toxicity, which configures an unfavorable microbiota [60,110]. Therefore, the reduction of urease from the consumption of oat porridge demonstrated an interesting prebiotic potential [60]. A more recent study with 210 mildly hypercholesterolemic individuals in China demonstrated a reduction in total and non-high-density lipoprotein levels after consuming 80 g of oats daily for 45 days [59]. Oat consumption also resulted in increased production of acetic acid and propionic acid when compared to the baseline, but not compared to the rice treatment (control) [59].

## 4. Conclusions and Future Perspectives

Current evidence suggests that oats can beneficially alter the composition of the gut microbiota and promote the synthesis of SCFAs. At present, most research has been conducted in animals and in vitro. These efforts suggest plausible mechanisms of action that demonstrate the need for future work in this area. Future clinical trials are needed to demonstrate the impact of oats on the human gut microbiota, with consideration for internal and external influences on the microbiota and high interindividual variability.

## Figures and Tables

**Figure 1 nutrients-15-03534-f001:**
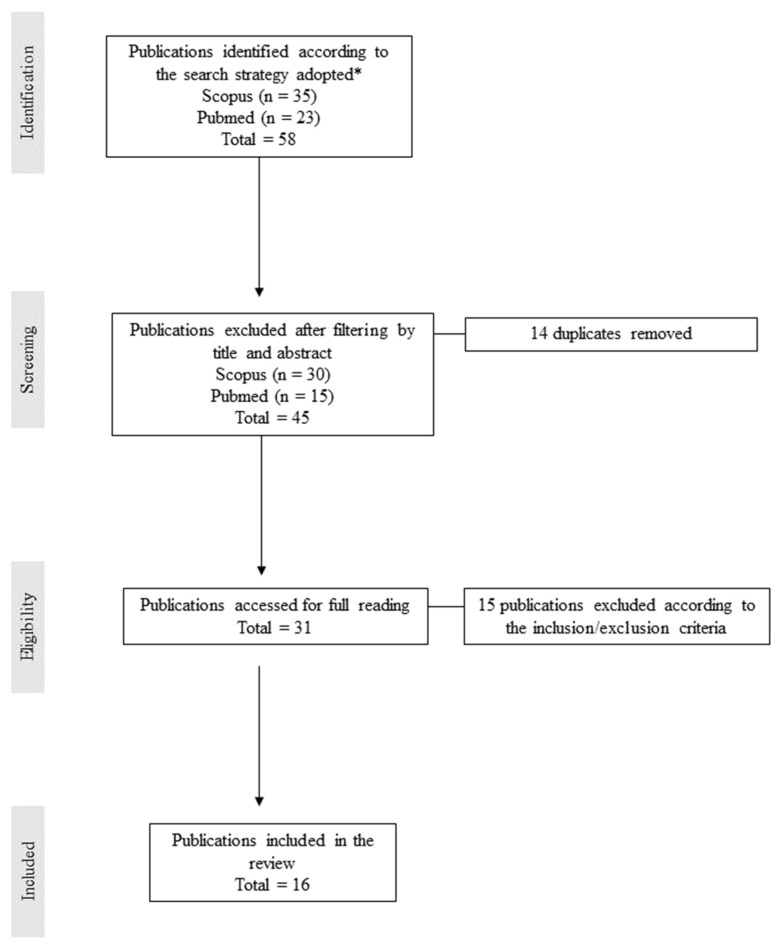
Methodology diagram of integrative review. * Advanced search by keywords “oat” and “microbiota” and “short-chain fatty acids” and “SCFA” (from 2012 to 2023).

**Figure 2 nutrients-15-03534-f002:**
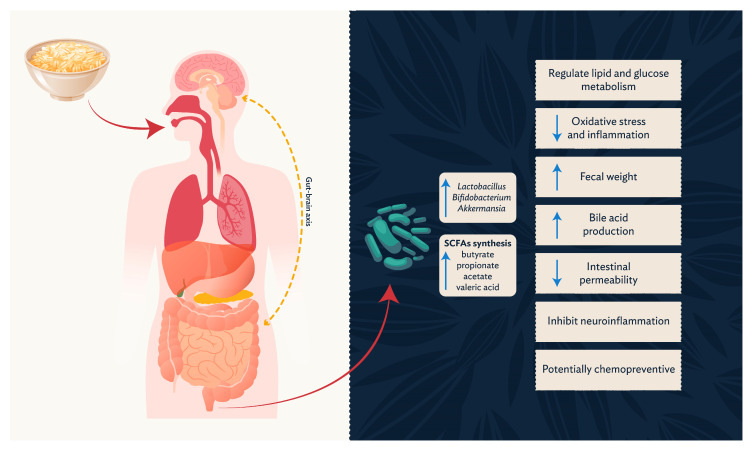
Effects of oat ingestion on gut modulation, including the increase of *Lactobacillus*, *Bifidobacterium* and *Akkermansia*. Further, oat intake results in increased butyrate, propionate, acetate, and valeric acid production. These microbes and metabolites are responsible for health benefits, such as regulation of lipid and glucose metabolism, decreases in oxidative stress and inflammation, increases in fecal weight, increased bile acid production, decreased intestinal permeability, inhibition of neuroinflammation, and potentially chemo-preventive properties. These systemic effects are possibly due to interactions within the gut–brain axis (dotted yellow arrow). The red arrows indicate the way of oat ingestion and digestion, and the blue arrows indicate an increase or decrease in markers of interest.

**Table 1 nutrients-15-03534-t001:** Scientific articles in vivo (animals) reporting the relation between oat consumption and microbiota modulation.

Authors (Year)	Country	Title	Journal	Food Matrix or Supplement	Aim	Methodology	Outcomes of Interest
Wang, Qi, Guo, Song, Pang, Fang, and Peng (2022) [46]	China	Effects of Oats, Tartary Buckwheat, and Foxtail Millet Supplementation on Lipid Metabolism, Oxide-Inflammatory Responses, Gut Microbiota, and Colonic SCFA Composition in High-Fat-Diet-Fed Rats	Nutrients	Cooked oats, tartaric buckwheat, and foxtail millet	Investigate the effect of cooked oats, tartaric buckwheat, and foxtail millet action on lipid levels, oxide-inflammatory responses, intestinal microbiota, and SCFA in rats.	Sixty male Sprague Dawley rats (n = 12 per group) were fed a basal diet, a high-fat diet (HFD), HFD with 22% cooked oats, HFD with 22% buckwheat, and HFD with 22% millet for 12 weeks.	Oats and buckwheat significantly decreased the expression of sterol regulatory element binding protein 2 and peroxisome proliferators activated γ receptors in liver tissue. Oats and buckwheat significantly increased *Lactobacillus* and *Romboutsia* in the microbiota. No impact of oats on alpha diversity of microbiota. Oats increased butyrate concentration by 2.16-times. Oats significantly decreased serum TNF-α, IL-6, and IL-1β. Oats and buckwheat significantly lowered MDA levels and increased SOD activities. Oats significantly reduced serum insulin and adiponectin.
Han, Gao, Song, Zhang, Li, and Zhang (2021) [47]	China	Oat Fiber Modulates Hepatic Circadian Clock via Promoting Gut Microbiota-Derived Short-Chain Fatty Acids	Journal of Agricultural and Food Chemistry	Oat Fiber	Evaluate the action of SCFAs produced by gut microbiota on circadian rhythm.	Seventy-two male C57BL/6 mice (24 per group) were fed a control diet, HFD, or HFD with 0.8% oat for 21 weeks.	Oat fiber significantly reduced the amount of weight gain from a HFD. Oat fiber significantly reduced total cholesterol, triglycerides, fasting insulin, and fasting glucose. Oat fiber increased production of acetate, propionate, and butyrate. Oat fiber reversed the effect of HFD on expression of circadian clock genes.
Ji, Ma, Zhang, Wang, Tao, Pei, and Hu (2021) [48]	China	Dietary Intake of Mixture Coarse Cereals Prevents Obesity by Altering the Gut Microbiota in High-Fat-Diet-Fed Mice	Food and Chemical Toxicology	Mix with millet, maize, oat, soybean, and purple potato	Evaluate the consumption of mixture coarse cereals on obesity prevention and gut microbiota in HFD-fed mice.	Forty-eight male C57BL/6 mice (n = 8 per group) were fed a chow diet (10% calories from fat) containing either 0%, 20%, or 40% mixture coarse cereals; or a HFD (45% calories from fat) containing either 0%, 20%, or 40% mixture coarse cereals for 8 weeks.	The dietary intake of mixture coarse cereals reduced body weight gain and fat accumulation in HFD-fed mice in a dose-dependent manner. High-dose cereal mixture decreased glycemic response in HFD-fed mice. Both doses of cereal mixture reduced total cholesterol and triglyceride levels in HFD-fed mice; only high-dose cereals reduced LDL-cholesterol in HFD-fed mice. High-dose cereal mixture reduced liver levels of TNF-α, IL-6, and IL-1β in HFD-fed mice. Both cereal mixtures decrease expression of hepatic fatty acid synthesis genes (acetyl-CoA carboxylase-1, fatty acid synthase, and SREBP-1c) in HFD-fed mice. Both cereal mixtures increased alpha diversity of gut microbiota in HFD-fed mice. High-dose cereal mixture reversed the increase of Firmicutes to Bacteroidetes ratio induced by HFD and increased Bacteroidetes and Actinobacteria phyla, as well as *Bifidobacterium*, *Holdemanella*, *Barnesiella*, *Okibacterium*, and *Streptophyta* genera in chow-fed mice but increased *Intestinimonas*, *Prevotella*, *Ruminococcus*, *Roseburia*, *Butyricimonas*, *Coprobacter*, *Bacteroides*, *Lactobacillus,* and *Turicibacter* genera, and *Lachnospiracea incertae sedis* species in HFD-fed mice. High-dose cereal mixture increased total SCFAs and acetic acid in colon of HFD-fed mice, but not to chow levels. There was no effect on propionate or butyrate.
Kundi, Lee, Pihlajamaki, Chan, Leung, Yu So, Nordlund, Kolehmainen, and El-Nezami (2020) [49]	China	Dietary Fiber from Oat and Rye Brans Ameliorate Western Diet–Induced Body Weight Gain and Hepatic Inflammation by the Modulation of Short-Chain Fatty Acids, Bile Acids, and Tryptophan Metabolism	Molecular Nutrition and Food Research	Oat bran	Elucidate the protective mechanisms conferred by oat and rye fibers in a Western diet.	Forty-eight male C57BL/6N mice (n = 12 per group) were fed a chow diet, a Western diet (WD), or a WD with 10% oat or rye bran, for 17 weeks.	Oat and rye brans reduced weight gain and ameliorated WD-induced glucose responses and the hepatic inflammation. Oat and rye reduced serum ALT and AST and attenuated hepatic inflammation. Oat reduced fasting blood glucose but had no effect on insulin. Oat reduced serum triglycerides but had no impact on LDL-cholesterol. Oat and rye increased expression of genes responsible for colonic mucin production and tight junctions. Oat and rye had no effect on liver enlargement or liver triglycerides. Oat and rye restored SCFA levels to that observed in chow-fed mice. Oat increased Bacteroidetes and Saccharibacteria phyla and *Lactobacillus* genus but reduced Firmicutes and Proteobacteria.
Gao, Song, Li, Zhang, Wan, Wang, Zhang, and Han (2020) [50]	China	Effects of Oat Fiber Intervention on Cognitive Behavior in LDLR−/− Mice Modeling Atherosclerosis by Targeting the Microbiome-Gut-Brain Axis	Journal of Agricultural and Food Chemistry	Oat fiber	Elucidate the oat fiber action on cognitive behavior through neuroinflammatory signals and the gut–brain axis.	Twenty male LDLR−/− mice (n = 10 per group) were fed a high-fat-and-cholesterol (HFC) diet (46% kcal from fat) with or without 0.8% oat fiber for 14 weeks. Ten male wild-type mice who were fed chow were used as controls.	HFC diet in LDLR−/− mice impaired spatial learning and memory compared to WT, which was restored by oat fiber in the HFC diet. Oat fiber reduced serum cholesterol, LDL-cholesterol, plaque formation, and neuroinflammation-related biomarkers in LDLR−/− mice on HFC diet. Oat fiber increased expression of SCFA receptors and tight junction proteins. Oat fiber increased total fecal SCFAs and acetate, propionate, and butyrate. Oat fiber increased abundance of microbiota and improved diversity by some indices (ACE, Chao, observed species), but not all indices (Shannon). Oat fiber decreased the ratio of Firmicutes and Bacteroidetes, as well as Proteobacteria and Bacteroidetes. Oat fiber increased abundance of Actinobacteria at the phylum level, Peptostreptococcaceae and Coriobacteriaceae at the family level, and *Eisenbergiella* and *Romboutsia* at the genus level. Oat fiber decreased abundance of Rikenellaceae at the family level, as well as *Anaerotruncus* and *Parabacteroides* at the genus level.
Huang, Yu, Li, Guan, Liu, Song, Liu, and Duan (2020) [51] *	China	Effect of Embryo-Remaining Oat Rice on the Lipid Profile and Intestinal Microbiota in High-Fat-Diet-Fed Rats	Food Research International	Embryo-remaining oat rice (EROR)	Investigate the effects of Embryo-remaining oat rice (EROR) on lipid profile, cecal SCFAs, and intestinal microbiota in HFD-fed rats.	Twenty-four male SD rats (n = 6 per group) were fed a normal diet, HFD, HFD with 10% EROR, HFD with 50% EROR for 4 weeks.	EROR at both levels did not impact body weight but blunted the elevation in serum total cholesterol, LDL-cholesterol, and triglycerides caused by HFD. EROR at both levels blunted the increase in ALT and AST caused by HFD and reduced expression of liver SREBP-1C, FAS, and HMGCR. High-dose EROR increased total SCFAs, acetate, and propionate, but not butyrate. EROR did not impact microbiota diversity (ACE, Shannnon) but increased the abundance of Actinobacteria phylum; Coriobacteriales, Bacillales, and Verrucomicrobiales order; Bifidobacteriaceae, Staphylococcaceae, and Coriobacteriaceae families; and *Akkermansia*, *Ruminococcus* spp., *Faecalibaculum* and *Roseburia* genera. EROR reduced the abundance of Firmicutes phylum and Clostridia class; Clostridiales and Lactobacillales orders, Lachnospiraceae and Lactobacillaceae families, and *Bacilli*, *Desulfovibrio*, *Holdemanella*, *Fusicatenibacter*, *Faecalitalea*, *Turicibacter*, and *Bifidobacterium* genera.
Sun, Tong, Liang, Wang, Liu, Zhou, and Zhou (2019) [52]	China	Effect of Oat and Tartary Buckwheat-based Food on Cholesterol-Lowering and Gut Microbiota in Hypercholesterolemic Hamsters	Journal of Oleo science	Oat (65%) and tartary buckwheat (25%) blend	Investigate the effects of oat-based food on cholesterol and the composition of the gut microbiota.	Thirty male golden hamsters (n = 10 per group) were fed a control diet, HFD, and HFD with 10% oat/buckwheat blend for 30 days.	Oat–buckwheat blend increased body weight but decreased plasma total cholesterol, LDL- cholesterol, liver total cholesterol, liver cholesterol ester liver triglycerides, compared to HFD. Oat–buckwheat blend increased fecal total cholesterol compared to HFD and control. Oat–buckwheat blend increased total SCFAs, acetate, propionate, and butyrate, compared to HFD and control. Oat–buckwheat blend decreased abundance of Bacteroidetes phylum; Erysipelotrichaceae, Ruminococcaceae, Lachnospiraceae, and Lactobacillaceae families; and *Ruminococcus* 1, *Ruminococcus* 2, and Ruminococcaceae-UCG-014 genera but increased the Eubacteriaceae family compared to HFD.
Dong, Zhu, Ma, Xiang, Shen, and Liu (2016) [53]	China	Oat Products Modulate the Gut Microbiota and Produce Anti-Obesity Effects in Obese Rats	Journal of Functional Foods	Oat meal: OM; oat flour: OF; and oat bran: OB	Compare the actions of oat meal, oat flour, and high-fiber oat bran on lipid metabolism, as well as gut microbiota of HFD-fed rats.	Eighty male SD (n = 10 per group) rats were fed either a chow-control diet, an HFD, an HFD with oat meal, an HFD with oat flour, or an HFD with high-fiber oat bran for 8 weeks.	Oat products attenuated HFD-induced increases in body weight and fat accumulation. Oat products reduced serum total cholesterol and triglycerides, with oat meal and oat bran having the greatest reductions compared to HFD. Oat products reduced serum endotoxin and TNF-α (serum levels and liver and adipose expression), and reduced adipocyte size with oat bran having the greatest reductions compared to HFD. Oat meal and oat bran resulted in lower abundance of Firmicutes and oat bran resulted in higher abundance of Bacteroidetes. Oat products did not impact microbiota diversity (Shannon, Richness Curve), but PCA suggests oat bran restores microbiota to a profile closer to the chow diet. Oat products increase total colonic SCFAs, acetate, and butyrate compared to HFD and chow diet; propionate increased compared to HFD only; oat bran resulted in greatest increases.
Wilczak, Blaszczyk, Kamola, Gajewska, Harasym, Jatosinska, Gudej, Suchecka, Oczkowski, Gromadzka–Ostrowska (2015) [54]	Poland	The Effect of Low- or High-Molecular-Weight Oat β-glucans on the Inflammatory and Oxidative Stress Status in the Colon of Rats with LPS-Indices Enteritis	Food and Function	Low- and high-molecular-weight oat β-glucans	Investigate the protective effect of low- and high-molecular-weight β-glucans in gut immune markers, microbiota changes and SCFAs.	Seventy-two male SD rats (n = 12 per group) were fed a control diet, low-molecular-weight β-glucan (1%) diet, or high-molecular-weight β-glucan (1%) diet for 6 weeks; half were administered LPS to induce enteritis.	In rats with enteritis, both β-glucans reduced IL-12 and IL-10 levels compared to control diet. In rats with enteritis, both β-glucans increased B lymphocytes in intraepithelial lymphocytes, but only high-molecular-weight β-glucan increased B lymphocytes in lamina propria lymphocytes compared to control diet; T lymphocytes were only increased by low-molecular-weight β-glucan in the lamina propria lymphocytes. In rats without enteritis, low-molecular-weight β-glucan increased T lymphocytes in intraepithelial lymphocytes and lamina propria lymphocytes compared to control diet. Both β-glucans increased antioxidant status in the colon compared to control diet in rats with and without enteritis. Both β-glucans increased abundance of lactic acid bacteria in feces compared to the control diet in rats with and without enteritis. In rats with enteritis, high-molecular-weight β-glucan increased acetate and propionate compared to the control diet; in rats without enteritis, only low-molecular-weight β-glucan increased propionate.

* The study used in vivo and in vitro experiments. Therefore, it is mentioned in this table and Table 2.

**Table 2 nutrients-15-03534-t002:** Scientific articles in vitro reporting the relation between oat consumption and microbiota modulation.

Authors (Year)	Country	Title	Journal	Food Matrix or Supplement	The Aim	Methodology	Outcomes of Interest
Bai, Zhang, Zhang, Zhang, Huo, Guo (2022) [55]	China	In Vitro Fermentation of Pretreated Oat Bran by Human Fecal Inoculum and Impact on Microbiota	Journal of Functional Foods	Oat bran pretreated	Determine the prebiotic effects of different pretreatments of oat bran.	Oat bran was steamed, microwaved, or dryed with hot air. The samples were exposed to in vitro simulated digestion and added to human fecal inoculum to investigate fermentation metabolites and impacts on microbiota.	Different heat-treatment processes did not impact β-glucan content. Steamed oat bran reduced *Escherichia*–*Shigella* and increased *Faecalibacterium prausnitzii*. Steamed oat bran generated more acetic, propionic, and butyric acid. Steamed oat bran had the greatest effect on gut microbiota diverstiy.
Liang, xie, Evivie, Zhao, Chen, Xu, Liu, Li, Huo (2021) [56]	China	Study on Supplementary Food with Beneficial Effects on the Gut Microbiota of Infants	Food Bioscience	Ready-to-use supplementary food with oat, corn, and millet	Investigate the impact of oat-, corn-, and millet-supplementary foods on gut microbiota and SCFA production.	Oat-, corn-, and millet-supplementary foods and an oligosaccharide control were added to human fecal inoculum from 6 infant donors (age 6–12 months) to investigate fermentation metabolites and impacts on microbiota.	Grain-based supplementary foods had a similar effect on microbiota diversity as oligosaccharides. Total SCFAs were similar across treatments, but the levels of propionate and butyrate produced by oat and corn supplementary foods were greater than oligosaccharides. The supplementary oat food increased the relative abundance of *Lactobacillus*.
Glei, Zetzmann, Lorkowschi, Dawczynski, and Scholormann (2020) [57]	Germany	Chemo-Preventive Effects of Raw and Roasted Oat Flakes After In Vitro Fermentation With Human Fecal Microbiota	International Journal of Food Sciences and Nutrition	Oat flakes	Analyzed the chemo-preventive effects of raw and roasted oat flakes, evaluating the processing in colon adenoma cells.	The oat flakes were roasted at 140 °C–160 °C for 20 min. The raw and roasted oat flakes were exposed to an in vitro simulated digestion and fermentation with human microbiota. Then, the fermentation supernatants (FS) obtained were characterized, and chemo-preventive effects were analyzed in LT97 colon adenoma cells.	During fermentation, the concentrations of SCFA, especially butyrate, were increased. Oat FS significantly decreased cell growth, which was time- and dose-dependent. The treatment with oat FS increased the caspase 3 activity, the expression of CAT, SOD2 and GSTP, while GPX1 mRNA was decreased. The study indicates chemo-preventive potential through growth inhibition and apoptosis, which is not affected by roasting.
Huang, Yu, Li, Guan, Liu, Song, Liu, and Duan (2020) [51] *	China	Effect of Embryo-Remaining Oat Rice on the Lipid Profile and Intestinal Microbiota in High-Fat-Diet-Fed Rats	Food Research International	Embryo-remaining oat rice (EROR)	Investigate the effects of an extract of EROR on lipid accumulation.	Water and lipid extracts of EROR were incubated with HepG2 cells to evaluate lipid accumulation in vitro.	Oat ethanol extracts significantly reduced lipid concentration, total cholesterol, and triglyceride in HepG2 cells.
Kristek, Wiese, Heuer, Kosik, Schar, Soycan, Alsharif, Kuhnle, Walton, and Spencer (2019) [58]	United Kingdom and Denmark	Oat Bran, But Not Its Isolated Bioactive β-glucans or Polyphenols, Has a Bifidogenic Effect in an In Vitro Fermentation Model of the Gut Microbiota	British Journal of Nutrition	Oat bran, β-glucan extract, oat polyphenols	Evaluate and compare the effects of an oatpolyphenol mix (avenanthramide, hydroxycinnamic acids, and benzoic acid derivatives), β-glucan extract (BG), and oat bran on gut microbiota.	The oat bran was digested in vitro, and the polyphenols were extracted from undigested (raw) and digested (after in vitro digestion) oat bran. Two different doses (1 and 3% (*w*/*v*)) of oat bran and matched concentrations of β-glucan extract or polyphenol mix were added to anaerobic fecal batch cultures.	Oat bran increased the abundance of Proteobacteria after 10 h and Bacteroidetes after 24 h, while the concentrations of acetic and propionic acids increased compared to the non-oat control. One percent oat bran increased the SCFA production at 24 h. It increased acetic and propionic acid, and increased the relative abundance of *Bifidobacterium*; polyphenols and isolated β-glucan did not impact SCFA production. The treatment with β-glucan induced an increase in Bacteroidetes at 24 h. The polyphenol mix induced an increase in the Enterobacteriaceae family after 24 h.
Carlson, Erickson, Hess, Gould and Slavin (2017) [4]	United States	Prebiotic Dietary Fiber and Gut Health: Comparing the In Vitro Fermentations of Beta-Glucan, Inulin, and Xylooligosaccharide	Nutrients	β-glucan and Oatwell (oat bran with 22% β-glucan)	Compare the fermentation effects of prebiotics (inulin, xylooligosaccharides and β-glucan based products) in the production of SCFA.	Five common prebiotic dietary fibers, OatWell, WholeFiber (dried chicory root blend with inulin, pectin, hemi and cellulose), xylooligosaccharide, pure inulin, and pure β-glucan, were incubated in fecal inoculum from healthy adults to measure microbiota, SCFA, and gas production.	Oatwell and β-glucan samples promoted the highest propionate at 24 h of fermentation. Inulin and WholeFiber increased *Collinsella* and resulted in the highest gas production at 12 h and 24 h of fermentation.

* The study used in vivo and in vitro experiments. Therefore, it is mentioned in Table 1 and this table.

**Table 3 nutrients-15-03534-t003:** Clinical studies reporting the relation between oat consumption and microbiota modulation.

Authors (Year)	Country	Title	Journal	Food Matrix or Supplement	The Aim	Methodology	Outcomes of Interest
Xu, Feng, Chu, Wang, Shete, Tuohy, Liu, Zhou, Kamil, Pan, Liu, Yang, Yang, Zhu, Lv, Xiong, Wang, Sun, Sun, and Yang (2021) [59]	China	The Prebiotic Effects of Oats on Blood Lipids, Gut Microbiota, and Short-Chain Fatty Acids in Mildly Hypercholesterolemic Subjects Compared with Rice: A Randomized, Controlled Trial	Frontiers in Immunology	Oat	Evaluate the relationship of blood lipids, intestinal microbiota, and SCFAs in a Chinese population with mild hypercholesterolemia.	In a randomized parallel design, 210 mildly hypercholesterolemic males and females from Beijing, Nanjing, and Shanghai were assigned to a diet containing 80 g/d of oats or rice (control) for 45 days, along with their habitual diet.	Total cholesterol and non-HDL cholesterol decreased in the oat group after 30 days and rice group after 45 days. The decrease in total cholesterol and non-HDL-cholesterol was greater in the oat group compared with the rice group at Day 45. Oat consumption significantly increased *Akkermansia muciniphila*, *Roseburia*, *Dialister*, *Butyrivibrio*, and *Paraprevotella*, and decreased Sutterellaceae. Oat consumption increased *Bifidobacterium* and was negatively correlated with LDL-C. In the oat group, total cholesterol and LDL-cholesterol were negatively correlated to *Faecalibacterium prausnitzii*. In the oat group, Enterobacteriaceae, *Roseburia*, and *Faecalibacterium prausnitzii* were positively correlated with plasma butyric acid and valeric acid concentrations and negatively correlated to isobutyric acid.
Valeur, Puaschitz, Midtvedt, and Berstad (2016) [60]	Norway	Oatmeal Porridge: Impact on Microflora-Associated Characteristics in Healthy Subjects	British Journal of Nutrition	Oatmeal porridge	Evaluate the effect of oatmeal porridge consumption every day for one week on fecal SCFAs, gas production, and inflammatory markers.	Ten healthy males and females consumed a daily portion of 60 g oatmeal porridge for 7 days. Lactulose-induced intestinal gas production, fecal excretion of SCFA, fecal levels of β-galactosidase and urease, and PGE2 levels were analyzed.	Fecal levels of β-galactosidase, urease, and PGE2 decreased after consumption of oatmeal porridge for one week.

## Data Availability

No new data were created or analyzed in this study. Data sharing is not applicable to this article.

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
