# Peer review of "Relationship between Oat Consumption, Gut Microbiota Modulation, and Short-Chain Fatty Acid Synthesis: An Integrative Review"

_nutrients, 2023, doi:10.3390/nu15163534_

Round 1

Reviewer 1 Report

In this manuscript, Relationship between oat consumption, gut microbiota modulation and short-chain fatty acids synthesis: an integrative review was studied well. But there are some questions in the aspects of designs, results and so on.

Hence, I have some suggestions as follows:

1) Some descriptions in the manuscript were not exact or confusing. Some words which will make the manuscript feel like an article on a popular science book should not appear in such a research paper. The following are suggestions for improving English usage. Please use standard expression in English. 

2) The title is not good, which is too general.

3) The manuscript stays within a stage of literature survey, and is hard to find original contribution of the authors on this subject.

 4) Problems on format or details: the manuscript was not well prepared according to the “Guidelines”. Please check carefully.

5) There is no clear description of treatment in "Methodology", so that it is very easy to create chaos in results. You had better transfer these general descriptions to special quantitative research.

6) If you put some photos into the paper, the design of your review will be more clearly understood.

 Extensive editing of English language required

Author Response

We would like to thank Reviewer 1 for reviewing our manuscript and providing valuable, constructive feedback. Detailed responses to reviewer comments follow as bullet points below the bolded initial comments with line numbers included when applicable. Revisions in the manuscript have been highlighted in yellow.

1) “Some descriptions in the manuscript were not exact or confusing. Some words which will make the manuscript feel like an article on a popular science book should not appear in such a research paper. The following are suggestions for improving English usage. Please use standard expression in English.”

  • Thank you for your comments to help us improve the readability of our manuscript. We have implemented these suggestions to the best of our ability and have asked for clarification when needed. We appreciate your feedback to enhance the clarity of our review.

2) “The title is not good, which is too general.

  • We appreciate this comment. Upon further review, we have left the title as is as we feel it encompasses the scope of our manuscript as our advanced search terms in completing our review included: “oat” AND “microbiota” AND “short chain fatty acids” AND “SCFA.” However, if you have further detailed suggestions to improve the title, we’d be happy to implement them.

3) “The manuscript stays within a stage of literature survey, and is hard to find original contribution of the authors on this subject.”

  • As a review article, we have developed this manuscript as a comprehensive summary of the current understanding of a modulation of the gut microbiota and its metabolites by oat consumption based on previously published research. Thus, while we discuss our inferences of the combined findings of previous research, original contributions have not been made beyond this.
  • If Reviewer 1 could point out which sections could benefit from original contributions and detail what they foresee this looking like, we would be happy to implement these changes upon our better understanding.

4) “Problems on format or details: the manuscript was not well prepared according to the “Guidelines”. Please check carefully.”

  • We have utilized Nutrients’ manuscript template and have updated formatting throughout to address this comment.

5) “There is no clear description of treatment in "Methodology", so that it is very easy to create chaos in results. You had better transfer these general descriptions to special quantitative research.”

  • Because the nature of this paper is an integrative review, we do not have a “treatment” to report.
  • We would like to point Reviewer 1 to lines 111-128 in the manuscript where the methodological approach utilized to complete the review is detailed, along with inclusion criteria, search terms, and quantitative statements on the number of articles initially obtained and remaining after screening. This process is further visualized in Figure 1.
  • However, if these points remain unclear upon our further explanation here and Reviewer 1 could expand upon their request to further clarify “treatments” and “quantitation” that would make the paper clearer, we’d be happy to revise accordingly.

6) “If you put some photos into the paper, the design of your review will be more clearly understood.”

  • We believe that Figure 1 accurately visualizes the design of our review. However, if Reviewer 1 has specifics on photos to include within the paper, we’d be happy to oblige.

Reviewer 2 Report

In the manuscript submitted to me for review entitled: „Relationship between oat consumption, gut microbiota modulation and short-chain fatty acids synthesis: an integrative reviewthe authors: Giovanna Alexandre Fabiano , Leila Marie Shinn and Adriane Elisabete Costa Antunes present an overview made on the basis of various studies presenting the benefit of the consumption of oats on the development of the intestinal microbiota and the metabolites that it releases as a result of its vital activity.

The research was done on the basis of 108 references, of which 47 are from the last 5 years (almost 1/2 of the total), 3 of which are from 2023. The main information is summarized from data obtained from more than 20 years. In the references presented, the authors do not use redundant self-citations.

My remarks and recommendations to the authors are:

1.     The beginning of line 222 begins with the inscription "Figure 2", which is in bold. Should it be so? This gives the impression that this is the inscription on the figure, and it actually starts from line 244.

2.     On line 320 the same note for "Table 3".

3.     My personal positive note to the authors. The tables are extremely well presented. It can be seen that the authors have put a lot of effort into it, and the result is an excellent presentation of the most important information included in the manuscript.

4.     After section 2. Methodology follows section 4. Oat as a functional food. Section number "3" is missing. Similarly, after section 4 follows section 6. Conclusion and future perspectives. Section number "5" is missing. Let the numbering of the sections in the manuscript be fixed.

Author Response

DETAILED RESPONSE TO COMMENTS BY REVIEWER #2

We would like to thank Reviewer 2 for reviewing our manuscript and providing positive feedback and diligent comments. Detailed responses to reviewer comments follow as bullet points below the bolded initial comments with line numbers included when applicable. Revisions in the manuscript have been highlighted in yellow.

1) “The beginning of line 222 begins with the inscription "Figure 2", which is in bold. Should it be so? This gives the impression that this is the inscription on the figure, and it actually starts from line 244.”

  • Thank you for your attention to detail. We have corrected this mistake (line 240)

2) “On line 320 the same note for "Table 3.

  • This modification was performed accordingly at lines 330 for Table 2 and line 338 for Table 3.

3) “My personal positive note to the authors. The tables are extremely well presented. It can be seen that the authors have put a lot of effort into it, and the result is an excellent presentation of the most important information included in the manuscript.

  • We appreciate this kind comment from Reviewer 2 as we worked diligently to ensure tables were as comprehensive and clear as possible.

4) “After section 2. Methodology follows section 4. Oat as a functional food. Section number "3" is missing. Similarly, after section 4 follows section 6. Conclusion and future perspectives. Section number "5" is missing. Let the numbering of the sections in the manuscript be fixed.

  • Again, we want to express their gratitude for Reviewer 2’s attention to detail and have corrected the section numbering throughout the manuscript.

Reviewer 3 Report

This is as good a review as any that I have previously read. It was a pleasure to read. 

Figure 1  "short chain fatty acids" and "SCFA"  I am assuming they are the same thing, so there is no need for "and".  

Line 164 :  concluded "that" 3 gm .......

I see no problems with Figure 2; it is actually  a good figure providing relevant information on the effects of oat ingestion.

Table 3 is not an easy to follow, but I believe that in the final version it will be much easier to follow. It is a big Table with a great deal of information. So that needs some attention to make it much easier to follow. 

The references are excellent and and provide a great deal of information from past work to very recent publications.

Other comments:  Was is the intention of the authors to concentrate on the benefits of oats, without considering some harmful aspects of consuming too much oats? (perhaps something to think about).

I would like to see the Title in a larger font, similar to the Names of the authors.

In all this is an excellent review, and should be published.

Author Response

DETAILED RESPONSE TO COMMENTS BY REVIEWER #3

We would like to thank Reviewer 3 for the favorable feedback and insightful suggestions. Detailed responses to reviewer comments follow as bullet points below the bolded initial comments with line numbers included when applicable. Revisions in the manuscript have been highlighted in yellow.

1) “Figure 1 "short chain fatty acids" and "SCFA" I am assuming they are the same thing, so there is no need for "and".”  

  • We understand your comment, but would like to clarify that we intended to include both the full name and abbreviation for short chain fatty acids in our three database searchers to ensure the most comprehensive article results possible. Thus, this statement in Figure 1 is noting that we used both the terms “short chain fatty acids” added to (AND) “SCFA” in our search methodology.

2) “Line 164 :  concluded "that" 3 gm .......”

  • Thank you for your attention to detail. We have corrected this mistake (line 180).

3) “I see no problems with Figure 2; it is actually a good figure providing relevant information on the effects of oat ingestion.”

  • We truly appreciate this positive feedback.

4) “Table 3 is not an easy to follow, but I believe that in the final version it will be much easier to follow. It is a big Table with a great deal of information. So that needs some attention to make it much easier to follow.”

  • Thank you for your comment. We understand that Table 3 has a great deal of information as we worked to create a comprehensive presentation of clinical studies reporting the relation between oat consumption and microbiota modulation for our readers. We have modified some formatting of the tables to improve the readability.

5) “The references are excellent and and provide a great deal of information from past work to very recent publications.”

  • Thank you for this affirmative comment.

6) “Other comments:  Was is the intention of the authors to concentrate on the benefits of oats, without considering some harmful aspects of consuming too much oats? (perhaps something to think about).”

  • Thank you for this insightful comment. We have addressed this concern expanding upon the AI of dietary fiber and lack of UL at lines 137-145: “According to the Food and Nutrition Board of the Institute of Medicine, the Ad-equate Intake (AI) for fiber is 14g of total fiber per 1000kcal, or 25g for adult women and 38g for adult men [46]. The AI for fiber represents the median fiber intake level observed to achieve the lowest risk of coronary heart disease [46]. While there is no Tolerable Upper Intake Level for fiber, excessive intake of dietary fiber can cause flatulence, bloating, and diarrhea. Therefore, consuming dietary fiber in adequate amounts can be beneficial to health, reducing the risk of several chronic diseases and supporting digestive health through modulation of laxation, fermentation, and the gut microbiota [46]. In this sense, oat intake can play a role in contributing to achieving the AI for fiber.”

7) “I would like to see the Title in a larger font, similar to the Names of the authors.”

  • We have made formatting modifications to improve the aesthetic of the manuscript, including increasing the Title and Names of the authors font sizes.

Round 2

Reviewer 1 Report

The draft of the paper has been partially revised, but it is hoped that the author can read through the full text again and revise it in detail, and the current version is basically acceptable.

The draft of the paper has been partially revised, but it is hoped that the author can read through the full text again and revise it in detail, and the current version is basically acceptable.